# Procalcitonin Level Monitoring in Antibiotic De-Escalation and Stewardship Program for Patients with Cancer and Febrile Neutropenia

**DOI:** 10.3390/cancers16203450

**Published:** 2024-10-11

**Authors:** Hiba Dagher, Anne-Marie Chaftari, Ray Hachem, Ying Jiang, Ann Philip, Patricia Mulanovich, Andrea Haddad, Peter Lamie, Rita Wilson Dib, Teny M. John, Natalie J. M. Dailey Garnes, Shahnoor Ali, Patrick Chaftari, Issam I. Raad

**Affiliations:** 1Department of Infectious Diseases, Infection Control, and Employee Health, The University of Texas MD Anderson Cancer Center, 1400 Pressler St., FCT12-6043, Unit 1460, Houston, TX 77030, USA; hrdagher@mdanderson.org (H.D.); rhachem@mdanderson.org (R.H.); yijiang@mdanderson.org (Y.J.); apphilip@mdanderson.org (A.P.); pmmulanovich@gmail.com (P.M.); ahaddad@metrohealth.org (A.H.); rita.wilsondib@ouhealth.com (R.W.D.); tmjohn1@mdanderson.org (T.M.J.); njdailey@mdanderson.org (N.J.M.D.G.); shahnoorali89@gmail.com (S.A.); iraad@mdanderson.org (I.I.R.); 2Department of Hospital Medicine, The University of Texas MD Anderson Cancer Center, Houston, TX 77030, USA; pmlamie@mdanderson.org; 3Department of Emergency Medicine, The University of Texas MD Anderson Cancer Center, Houston, TX 77030, USA; pchaftari@mdanderson.org

**Keywords:** procalcitonin, biomarker, cancer, neutropenia, fever, febrile neutropenia

## Abstract

**Simple Summary:**

Procalcitonin (PCT) is a blood biomarker that can be used to detect infections and is often combined with clinical judgment to guide antibiotic use, particularly in critically ill patients and those with respiratory infections. In this study, we aimed to evaluate how PCT levels can help guide antibiotic treatment in cancer patients with febrile neutropenia. We found that a 30% decrease in PCT levels or a repeated PCT level of ≤ 0.25 ng/mL was associated with earlier reduction of antibiotics and shorter treatment duration, without affecting patient outcomes. This suggests that monitoring PCT could safely and effectively optimize antibiotic use in these patients, reducing the risk of antibiotic resistance.

**Abstract:**

Objective: Serial procalcitonin (PCT) monitoring has been adopted to supplement clinical judgement and help guide antibiotic therapy as part of antimicrobial stewardship programs. PCT levels peak 24 to 48 h after infection onset and decline with infection resolution. We explored the role of PCT as an infection biomarker for guiding antibiotic therapy in cancer patients hospitalized for febrile neutropenia. Design: Prospective randomized study. Methods: Patients were enrolled between October 2021 and August 2023 and received empiric intravenous broad-spectrum antibiotics (IVBSA) for at least 48 h. PCT was measured at baseline and 48–72 h after IVBSA initiation. PCT drop 48–72 h after IVBSA initiation was defined as a reduction of 30% from baseline or a PCT level < 0.25 ng/mL. De-escalation was defined as a switch from IVBSA to oral or simplified once-daily IV therapy. Results: Of the 89 patients with available PCT levels, 53 (60%) had a PCT drop, most of whom (79%) underwent IVBSA de-escalation. Compared with patients without a PCT drop, patients with a PCT drop had a higher de-escalation rate at 72 h (71% vs. 45%; *p* = 0.003) and a shorter median antibiotic duration (55 h vs. 98 h; *p* = 0.004). Patients with bacteremia had a significantly higher median PCT level than those without bacteremia (2.35 ng/mL vs. 0.370 ng/mL, *p* = 0.013). Conclusions: In patients with cancer and febrile neutropenia, a PCT drop was associated with earlier therapy de-escalation and shorter antibiotic duration. PCT monitoring may be useful in antimicrobial stewardship initiatives in this patient population. Clinical trials identifier: NCT04983901.

## 1. Introduction

Procalcitonin (PCT) level monitoring has been used to detect bacterial infections and sepsis; a high PCT level suggests a bacterial etiology [1]. The PCT level typically peaks 24 to 48 h after the onset of an infection and gradually declines as the infection resolves [2,3,4,5,6,7]. Serial monitoring of PCT level has been recently incorporated into clinical algorithms, particularly into those used for patients with respiratory infections, to complement clinical judgment and help guide antibiotic therapy, improve patient outcomes, support antimicrobial stewardship initiatives, and decrease antibiotic resistance [8,9,10,11,12]. In one study, patients in intensive care units (ICU) who were treated according to a PCT-guided algorithm with a PCT threshold of 0.5 ng/mL had a lower antibiotic exposure and shorter antibiotic duration, while achieving similar clinical outcomes compared to those not treated according to the algorithm [8,10]. Similar results were observed in ICU patients in the Netherlands and in patients with lower respiratory infections in Switzerland [12,13]. However, in the US, using similar PCT thresholds did not result in reduced antibiotic use in patients with suspected lower respiratory infections [14]. In addition, when a lower PCT threshold (0.1 ng/mL) was used in an algorithm for critically ill patients in another study, it did not indicate reduced antibiotic exposure, but it was a good predictor of survival [11]. Hence, there remains uncertainty regarding the optimal PCT threshold for guiding antibiotic use.

In patients with cancer and febrile neutropenia, a PCT threshold of 0.25 ng/mL was shown to accurately predict bloodstream infection, hospital admission, and prolonged length of stay [15]. In this prospective study, we evaluated PCT kinetics and explored the potential of PCT to guide antibiotic therapy de-escalation, determine antibiotic therapy duration, and predict clinical and microbiological outcomes and adverse events.

## 2. Methods

### 2.1. Study Design

Between 7 October 2021, and 31 August 2023, we prospectively enrolled patients with cancer and febrile neutropenia who required intravenous broad-spectrum antibiotics (IVBSA) and randomized them to receive either the combination of imipenem/cilastatin/relebactam or standard-of-care antibiotics consisting of cefepime, piperacillin-tazobactam, or meropenem for at least 48 h. Patients in both arms were allowed to also receive antibiotics targeting gram-positive bacteria (IV vancomycin, linezolid, or daptomycin) as clinically indicated. This was not a randomized trial of the biomarker PCT with a clinical decision algorithm concerning the de-escalation of IVBSA.

This study was registered at ClinicalTrials.gov (NCT04983901), approved by the Institutional Review Board of The University of Texas MD Anderson Cancer Center, and funded by Merck & Co. Written informed consent was obtained from all patients or their authorized representatives prior to conducting any study-related procedures.

The trial’s safety and efficacy data are presented in a different paper [16].

### 2.2. PCT Level and Outcomes

Patients’ PCT levels were measured at baseline (within 24 h of enrollment) and 48 to 72 h after IVBSA initiation. A PCT drop 48 to 72 h after starting IVBSA was defined as a 30% decrease from the baseline PCT level or a PCT level below 0.25 ng/mL. De-escalation of antibiotics was defined as the transition from IVBSA to either oral antibiotics or a convenient once-daily IV regimen. De-escalation was left to the discretion of the primary medical team taking care of the patients and was mainly guided by each patient’s clinical response and not based on a PCT-guided algorithm. Patients were followed during IVBSA administration and for 42 days thereafter, and were monitored for clinical and microbiological outcomes, antibiotic therapy de-escalation, antibiotic therapy duration, and the occurrence of adverse events. A favorable clinical outcome was defined as the resolution of all acute signs and symptoms related to the primary infection, primarily evaluated by fever resolution. In contrast, clinical failure was identified as the ongoing presence or worsening of signs and symptoms during treatment. A favorable microbiological outcome was defined as repeat negative cultures in cases of documented infections, or presumed culture eradication in cases where there was a favorable clinical response without further cultures. Conversely, microbiological failure was defined as the persistence of positive cultures with the baseline pathogen, or presumed persistence in instances of clinical failure without repeat cultures.

### 2.3. Statistical Analysis

Categorical variables were compared using the chi-square or Fisher exact test, as appropriate. Continuous variables were compared using the Wilcoxon rank-sum test. In addition, for patients who experienced de-escalation of antibiotic therapy, the Cochran–Armitage trend test was used to analyze trends between patients’ PCT drop and different durations of antibiotic use. All tests were two-sided and had a significance level of 0.05. Data analyses were performed using SAS version 9.4 (SAS Institute Inc., Cary, NC, USA).

## 3. Results

### 3.1. Patients and PCT Kinetics

We enrolled 100 patients in the clinical trial of IVBSA, of whom we identified 89 patients who had available PCT levels. Of those, 53 patients (60%) had a PCT drop of at least 30% from baseline or a PCT level of less than 0.25 ng/mL 48 to 72 h after IVBSA initiation. Patients’ characteristics were similar among patients who experienced a PCT drop vs. those who did not (Table 1). Although the difference was not significant, patients who experienced a PCT drop were more likely to have received IMI/REL than those who did not experience a PCT drop (60% vs. 42%; *p* = 0.08).

### 3.2. De-Escalation of Antibiotic Therapy

Of the 53 patients who had a PCT drop, 42 (79%) underwent de-escalation of IVBSA (Table 2).

Among the 64 patients who underwent IVBSA de-escalation, those who experienced a PCT drop had a significantly shorter median antibiotic therapy duration than those who did not (55 h vs. 92 h; *p* = 0.004).

The de-escalation of antibiotic therapy occurred sooner in patients who experienced a PCT drop than in the those who did not (*p* = 0.003). Such a de-escalation was observed within 72 h in 71% of the patients who experienced a PCT drop versus in 45% of those who did not (Table 2).

Patients who experienced a PCT drop and those who did not had similar clinical and microbiological outcomes at the end of IVBSA therapy (Table 2). Patients who experienced a PCT drop and those who did not also did not have significantly different rates of adverse events (*p* = 0.67), particularly regarding the occurrence of infections after the end of IVBSA therapy.

Patients with documented bacteremia exhibited a significantly higher median PCT level within 24 h of baseline than those without bacteremia (median PCT of 2.35 ng/mL vs. 0.370 ng/mL; *p* = 0.013) (Figure 1).

## 4. Discussion

We found that in patients with cancer who had febrile neutropenia, a PCT drop of 30% from baseline PCT level or a PCT level of less than 0.25 ng/mL was associated with earlier IVBSA de-escalation and shorter antibiotic duration that did not impact overall clinical outcome.

Our findings are consistent with those of previous studies that showed that PCT level could be used to guide clinicians’ decisions to discontinue antibiotics, leading to decreased treatment duration and antibiotic exposure without adversely affecting the clinical outcome [12,17]. However, unlike our study, these studies were conducted in critically ill patients, used a higher PCT threshold (0.5 ng/mL), and defined a PCT drop as a bigger reduction (80%) from its peak value [18]. Other studies involving subjects with respiratory infections and critically ill subjects used a lower threshold of 0.25 ng/mL and had similar results [9,19,20]. In the present study, we explored using a lower PCT threshold to guide antibiotic de-escalation as a cautious strategy to ensure the safety of patients with cancer who are immunocompromised. We selected a threshold of 0.25 ng/mL on the basis of a previous study that showed that a PCT level of at least 0.25 ng/mL in this patient population strongly predicted bacteremia [15]. Furthermore, a study in patients with community-acquired pneumonia found that PCT was a good predictor of bacteremia, and that patients with a PCT level of less than 0.25 ng/mL had a low risk (<1%) of bacteremia [21]. We chose to define PCT drop using a smaller value of 30% rather than the previously reported 80% because patients with cancer may present with an elevated PCT level owing to their underlying malignancy, which could further increase in those with bacteremia [15]. Another study showed that PCT was higher in cancer patients compared to non-cancer patients and even more elevated in febrile cancer patients with sepsis or bacteremia [22]. Therefore, PCT may increase in patients with cancer, not solely as a result of infection, but also as a consequence of their underlying malignancy. As a result, cancer patients may not experience the same magnitude of PCT decline as non-cancer patients, where infection is often the only factor driving PCT elevation. Furthermore, our choice of 30% is supported by another study that demonstrated that cancer patients with bacterial infections who responded to antimicrobial therapy had a 30% decrease in their median PCT levels [23].

In the present study, PCT level declined in 60% of the patients after starting IVBSA. PCT level has been used to monitor the response to antibiotic therapy [3]. In patients with cancer who have bacterial infections, a decrease in PCT level has been shown to reflect the response to antimicrobial therapy, with a significant decline in PCT level of 30% in patients who responded to therapy compared with those who did not [23]. Previous studies have also shown a correlation between PCT level and infection severity, with patients who have sepsis or septic shock having higher PCT levels [3,4,24]. In the present study, we observed that in patients with bacteremia, the median PCT level was significantly higher (2.35 ng/mL) than that in those without bacteremia (0.370 ng/mL); these findings are similar to those of a previous study of patients with neutropenia, which showed that PCT levels were significantly higher in patients with bacteremia than in those without a documented infection [23]. Similarly, in a study of patients with community-acquired pneumonia, PCT level was a good predictor of bacteremia [21]. Another study in adult patients with fever showed that a PCT level below 0.4 ng/mL accurately ruled out bacteremia with a sensitivity rate of 95% and a negative predictive value of 98.8% [25].

We found an association between the PCT drop of 30% and the antibiotic treatment duration. Although the decision to de-escalate antibiotic therapy was not based on a PCT-guided algorithm, but left at the discretion of the treating physician primarily based on clinical judgment, it is possible that the clinician’s awareness of the PCT levels may have influenced the decision to de-escalate or interrupt antibiotic treatment. The clinician ‘s unblinded access to the PCT results may have introduced bias, potentially impacting their clinical decisions. The small sample size and the specific patient population may limit the generalizability of our findings to a wider patient population. In addition, our study included patients with various types of malignancies, who may have different degrees of immunosuppression despite the profound neutropenia at presentation, underlying the heterogeneity of our patient population. Future larger studies with subgroup analyses are warranted to validate these findings. Given that PCT levels can be influenced by both underlying malignancy and inflammatory response, distinguishing between infection-related and tumor-related PCT elevations can be challenging. Therefore, caution is advised when interpreting PCT results in oncological patients. Additionally, we chose a 30% PCT drop, which is lower than the commonly referenced 80% drop in other populations. This choice was based on evidence suggesting that cancer patients often have elevated baseline PCT levels due to their malignancy, even in the absence of infection [22]. Although our choice of 30% is supported by a study showing that a 30% PCT decrease reflects a response to antimicrobial therapy in cancer patients with bacterial infections [23], further research is needed to validate the appropriateness of utilizing this drop and to better understand the PCT dynamics in this population. Moreover, the use of a lower threshold may limit the generalizability of our findings to non-cancer populations, in whom a greater PCT decline is typically observed. Another limitation of this study is that the repeat PCT levels were not assessed consistently at the same timepoint in all patients which may have introduced some potential biases in the interpretation of the results.

## 5. Conclusions

In summary, most patients who experienced a PCT drop underwent antibiotic de-escalation sooner than patients who did not experience a PCT drop. Hence, patients who experienced a PCT drop received a shorter course of antibiotics without negatively impacting their clinical outcome. Although we do not recommend using a single PCT measurement to diagnose sepsis or bacterial infections, using a combination of serial PCT monitoring that has been incorporated into a clinical algorithm and clinical judgement may decrease antibiotic exposure without negatively impacting the clinical outcome in patients with cancer who are the most vulnerable. PCT monitoring may also be useful in antimicrobial stewardship initiatives and may guide providers in the judicious use of anti-microbial agents in this population. A large prospective randomized study of cancer patients to evaluate the role of PCT incorporated in a clinical algorithm is warranted.

## Figures and Tables

**Figure 1 cancers-16-03450-f001:**
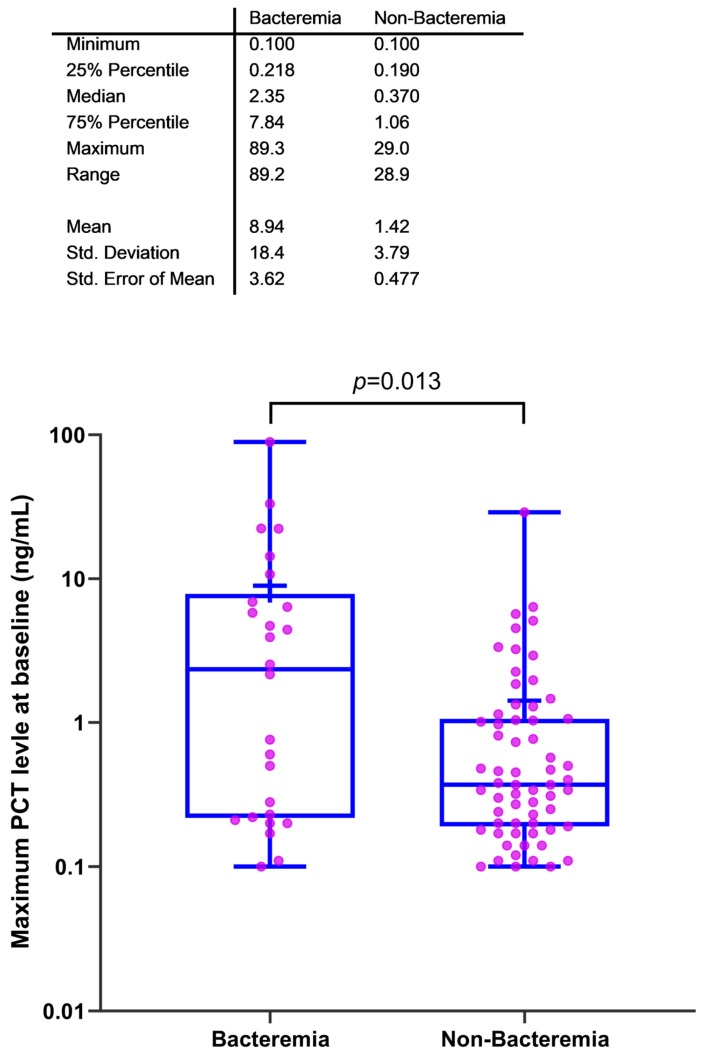
Boxplot for baseline PCT levels in patients with and without bacteremia.

**Table 1 cancers-16-03450-t001:** Descriptive statistics of patients in this study.

Characteristics	Patients	PCT Drop ≥ 30% or PCT < 0.25 ng/mL at 48–72 h		
	Total	No	Yes	*p*-Value
	(*n* = 89)	(*n* = 36)	(*n* = 53)	
	*n* (%)	*n* (%)	*n* (%)	
Age (years), median (range)	58 (18–84)	58 (18–84)	58 (24–80)	0.84
Gender, male	50 (56)	24 (67)	26 (49)	0.10
Race				0.22
White	62 (70)	30 (83)	32 (60)	
Black	8 (9)	2 (6)	6 (11)	
Hispanic	12 (13)	2 (6)	10 (19)	
Asian	4 (4)	1 (3)	3 (6)	
Middle Eastern	2 (2)	1 (3)	1 (2)	
Other	1 (1)	0 (0)	1 (2)	
Type of cancer				0.74
Hematological malignancy	60 (67)	25 (69)	35 (66)	
Solid tumor	29 (33)	11 (31)	18 (34)	
BMT within one year	8 (9)	3 (8)	5 (9)	>0.99
Type of BMT				0.46
Autologous	3/8 (38)	2/3 (67)	1/5 (20)	
Allogeneic	5/8 (62)	1/3 (33)	4/5 (80)	
GVHD	3/8 (38)	1/3 (33)	2/5 (40)	>0.99
Positive blood culture (Bacteremia)	26 (29)	12 (33)	14 * (26)	0.48
*Escherichia coli*	5	3	2	
*Pseudomonas aeruginosa*	4	2	2	
*Enterobacter cloacae*	2	1	1	
*Klebsiella pneumoniae*	1	1		
*Haemophilus parainfluenzae*	1		1	
*Rhizobium radiobacter*	1	1		
*Roseomonas mucosa*	1		1	
*Streptococcus mitis/oralis Group*	8	3	5	
*Staphylococcus epidermidis*	3		3	
*Enterococcus faecalis*	2		2	
*Enterococcus faecium*	1	1		
*Micrococcus luteus*	1		1	
*Rothia mucilaginosa*	1		1	
*Dermacoccus nishinomiyaensis*	1		1	
IVBSA treatment				0.08
Imipenem/Cilastatin/Relebactam	47 (53)	15 (42)	32 (60)	
Standard of Care (SOC)	42 (47)	21 (58)	21 (40)	

BMT, bone marrow transplant; GVHD, graft-versus-host disease; IVBSA, intravenous broad-spectrum antibiotics; PCT, procalcitonin. All values are *n* (%) unless otherwise indicated. * Note: A few patients had multiple pathogen species (2–3) identified from their blood cultures.

**Table 2 cancers-16-03450-t002:** Association analysis of PCT change with outcomes.

Outcomes	PCT Drop ≥ 30% or PCT < 0.25 ng/mL at 48–72 h		*p*-Value
	No	Yes	
	(*n* = 36)	(*n* = 53)	
	*n* (%)	*n* (%)	
Antibiotic therapy de-escalation to oral or once-daily IV therapy	22 (61)	42 (79)	0.06
For patients with antibiotic therapy de-escalation	*n* = 22	*n* = 42	
Duration of antibiotic use (hours), median (IQR)	92 (58–121)	55 (42–73)	0.004
For patients with antibiotic therapy de-escalation	*n* = 22	*n* = 42	
Duration of antibiotic use			0.003 *
< = 72 h	10 (45)	30 (71)	
72–96 h	3 (14)	7 (17)	
96–120 h	2 (9)	2 (5)	
120–168 h	3 (14)	3 (7)	
> 168 h	4 (18)	0 (0)	
Clinical outcome at EOIV			0.12
Favorable clinical response	26 (72)	46 (87)	
Clinical failure	9 (25)	7 (13)	
Indeterminate	1 (3)	0 (0)	
For patients with microbiology documentation	*n* = 12	*n* = 15	
Microbiology response at EOIV			0.70
Eradication	11 (92)	14 (93)	
Presumed eradication	0 (0)	1 (7)	
Indeterminate	1 (8)	0 (0)	
Adverse events (AE)—New infections	20 (56)	27 (51)	0.67
Any AE	25 (69)	34 (64)	0.60
Any SAE	24 (67)	30 (57)	0.34
Study-drug-related AE	5 (14)	5 (9)	0.52
Study-drug-related SAE	1 (3)	1 (2)	>0.99

Note: * The Cochran–Armitage trend test was used for the analysis. AE, adverse event; EOIV, end of intravenous therapy; IV, intravenous; IQR, interquartile range; PCT, procalcitonin, SAE, serious adverse event. All values are *n* (%) unless otherwise indicated.

## Data Availability

Data supporting the reported results in this study are not publicly available due to HIPAA regulations and the need to protect patient confidentiality. However, access to de-identified data may be requested from the corresponding author under reasonable conditions, and will be considered in accordance with institutional and legal guidelines.

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
