# Peer review of "Procalcitonin Level Monitoring in Antibiotic De-Escalation and Stewardship Program for Patients with Cancer and Febrile Neutropenia"

_cancers, 2024, doi:10.3390/cancers16203450_

Round 1

Reviewer 1 Report

Comments and Suggestions for Authors

The authors attempted to investigate whether the drop of 30% from baseline PCT level or a PCT level of less than 0.25 ng/mL predicts the clinical outcome and influences antibiotic therapy duration. 

Authors suggest that PCT threshold at 0.25 ng/mL can be safer for patients with febrile neutropenia as a result of antineoplastic treatment, which seems to be valuable observation worth checking in clinical practice.

Although analysis brought some important results, there are few limitations of the study: 

- low number of patients 

- variety of diagnoses - febrile neutropenia and sepsis in patients with hematologic malignancies, especially in those who have undergone BMT is not comparable to those treated with  solid tumors, where the bone marrow is not primarily involved

- non repeatable time points for pct evaluation 

- non specific increase in PCT concentration due to ongoing malignancy can be difficult to distinguish from inflammatory process in some cases

In my opinion,  the group is to small and heterogenous to obtain objective data, but analysis is worth to be extended  by a larger group of patients and can have important clinical meaning.

Author Response

The authors attempted to investigate whether the drop of 30% from baseline PCT level or a PCT level of less than 0.25 ng/mL predicts the clinical outcome and influences antibiotic therapy duration.

Authors suggest that PCT threshold at 0.25 ng/mL can be safer for patients with febrile neutropenia as a result of antineoplastic treatment, which seems to be valuable observation worth checking in clinical practice.

Although analysis brought some important results, there are few limitations of the study:

- low number of patients

- variety of diagnoses - febrile neutropenia and sepsis in patients with hematologic malignancies, especially in those who have undergone BMT is not comparable to those treated with  solid tumors, where the bone marrow is not primarily involved

- non repeatable time points for pct evaluation

- non specific increase in PCT concentration due to ongoing malignancy can be difficult to distinguish from inflammatory process in some cases

In my opinion,  the group is to small and heterogenous to obtain objective data, but analysis is worth to be extended  by a larger group of patients and can have important clinical meaning.

Response: Thank you for your detailed feedback on our manuscript. We appreciate the opportunity to address the points you raised and to clarify the limitations of our study.

- low number of patients: we agree with you that the sample size in our study is relatively small, which does limit the generalizability of the findings. We have acknowledged this limitation in the “Discussion” section in the manuscript. As you suggested, we have added that future larger studies are warranted to validate these findings.

- variety of diagnoses: You are correct in highlighting the heterogeneity of the patient population, particularly between those with hematologic malignancies (undergoing BMT) and those with solid tumors. Despite the fact that all these patients were profoundly neutropenic, we recognize that these groups may differ significantly in terms of bone marrow involvement and immune response. We have included this limitation in the discussion section. This will be a great idea to pursue in future larger studies, where we can perform subgroup analyses.

- non repeatable time points for pct evaluation: We agree with the reviewer and have acknowledged this limitation in the discussion section: “the repeat PCT levels were not done consistently at the same timepoint in all patients which may have introduced some potential biases in the interpretation of the results”.

- non specific increase in PCT concentration due to ongoing malignancy can be difficult to distinguish from inflammatory process in some cases: We agree that PCT levels can be influenced by both the underlying malignancy and the inflammatory response, which may sometimes make it difficult to distinguish between infection-related and tumor-related PCT elevation. A previous study has shown that PCT was higher in cancer patients compared to non-cancer patients and was even higher in febrile cancer patients with sepsis or bacteremia.  This reference was added to the manuscript. However to address this reviewer’s concern, we have emphasized in the discussion section that PCT should be interpreted with caution, especially in oncology patients.

In my opinion,  the group is to small and heterogenous to obtain objective data, but analysis is worth to be extended  by a larger group of patients and can have important clinical meaning. We agree with the reviewer and have added in the discussion that future larger studies with subgroup analyses are warranted to validate our findings.

Reviewer 2 Report

Comments and Suggestions for Authors

There is one major problem with this article, namely that it may unintentionally give the impression that it is a randomized trial of the biomarker procalcitonin with a clinical decision algorithm concerning the de-escalation of IVABT.

The authors should clearly emphasize in the methods section that this is not the case and not just point out in the discussion that they do not know to what extent the procalcitonin results have influenced any treatment decision (“not guided by PCT but left at the discretion…”)

The following information is missing from this article

·        Total number of study patients compared to those who received a procalcitonin measurement.

·        The number and proportion of patients with a positive blood culture (calculated from Figure 1 26 of 89, which would correspond to a high positive rate of 29 %)

·        Distribution of pathogen species in the positive blood cultures. This may be an explanation for different response rates to the first line empiric treatment.

·        If all these data are “reserved” for the main publication, it should be published first so that it can be referenced.

Author Response

There is one major problem with this article, namely that it may unintentionally give the impression that it is a randomized trial of the biomarker procalcitonin with a clinical decision algorithm concerning the de-escalation of IVABT.

The authors should clearly emphasize in the methods section that this is not the case and not just point out in the discussion that they do not know to what extent the procalcitonin results have influenced any treatment decision (“not guided by PCT but left at the discretion…”)

The following information is missing from this article

  • Total number of study patients compared to those who received a procalcitonin measurement.
  • The number and proportion of patients with a positive blood culture (calculated from Figure 1 → 26 of 89, which would correspond to a high positive rate of 29 %)
  • Distribution of pathogen species in the positive blood cultures. This may be an explanation for different response rates to the first line empiric treatment.
  • If all these data are “reserved” for the main publication, it should be published first so that it can be referenced.

Response: Thank you for your valuable feedback on our manuscript. We appreciate your thoughtful suggestions and have addressed each point as follows:

- Clarification of the study design and methods:

We understand your concern that the article may unintentionally give the impression of being a randomized trial of the biomarker procalcitonin with a clinical decision algorithm for IV antibiotic therapy de-escalation. We apologize for this confusion. To clarify this, and as per your suggestion, we have now emphasized in the Methods section that this was not a randomized trial of the biomarker PCT with a clinical decision algorithm concerning de-escalation of IVBSA. We explicitly state that treatment decisions were mainly guided by patient’s clinical response and not based on a PCT-guided algorithm but were left to the discretion of the treating physician (Methods sections 2.1 and 2.2). We believe this revision will help eliminate any potential misinterpretation of the study design.

  • Total number of study patients compared to those who received a procalcitonin measurement:

We have added the total number of study patients compared to those who received a procalcitonin measurement to the Methods section for clarity. We enrolled 100 patients in the clinical trial of IVBSA, of whom we identified 89 patients who had available PCT levels.

  • The number and proportion of patients with a positive blood culture (calculated from Figure 1 → 26 of 89, which would correspond to a high positive rate of 29 %)

The number and proportion of patients with positive blood cultures, as you calculated from Figure 1 (26 of 89, corresponding to 29%), have now been included in the Table 1, with track changes and highlighted in blue color.

  • Distribution of pathogen species in the positive blood cultures. This may be an explanation for different response rates to the first line empiric treatment.

The distribution of pathogen species in positive blood cultures have been included in the Table 1 of the revised manuscript.

  • If all these data are “reserved” for the main publication, it should be published first so that it can be referenced.

The main publication has been published and the reference has been added to this manuscript (new reference #16).

Reviewer 3 Report

Comments and Suggestions for Authors

Thank you for the opportunity to review this manuscript describing a prospective, randomized trial in adult cancer patients with febrile neutropenia (1) comparing the safety and efficacy of imipenem-relebactam with the standard-of-care treatment (cefepime, meropenem, or piperacillin/ tazobactam), and (2) investigating the utility of PCT monitoring in guiding antibiotic de-escalation and duration of use.

1. The Study Design is sound. Your decision to use a 30% drop in PCT or PCT <0.25 ng/mL is supported by previous studies, as detailed in the Discussion.

2. The descriptive statistics and statistical methods are appropriate, and the results clearly presented in Table 1 and Figure 1.

3. The Discussion is outstanding. You have compared your results to previous studies, emphasizing the fact that most of the previous evidence was obtained from critically ill, often septic patients, utilizing more aggressive definitions of a drop in PCT. Your discussion of study limitations is appropriate.

4. The conclusions are evidence based and appropriate.

Good luck with your manuscript.

Author Response

Thank you for the opportunity to review this manuscript describing a prospective, randomized trial in adult cancer patients with febrile neutropenia (1) comparing the safety and efficacy of imipenem-relebactam with the standard-of-care treatment (cefepime, meropenem, or piperacillin/ tazobactam), and (2) investigating the utility of PCT monitoring in guiding antibiotic de-escalation and duration of use.

  1. The Study Design is sound. Your decision to use a 30% drop in PCT or PCT <0.25 ng/mL is supported by previous studies, as detailed in the Discussion.
  2. The descriptive statistics and statistical methods are appropriate, and the results clearly presented in Table 1 and Figure 1.
  3. The Discussion is outstanding. You have compared your results to previous studies, emphasizing the fact that most of the previous evidence was obtained from critically ill, often septic patients, utilizing more aggressive definitions of a drop in PCT. Your discussion of study limitations is appropriate.
  4. The conclusions are evidence based and appropriate.

Good luck with your manuscript.

Response: Thank you for your time to thoroughly review our manuscript and for your positive feedback. We hope that our work will also be of significant interest to the healthcare professionals and researchers caring for the vulnerable immunocompromised cancer patients.

Reviewer 4 Report

Comments and Suggestions for Authors

I reviewed the manuscript entitled "Procalcitonin level monitoring in antibiotic de-escalation and stewardship program in patients with cancer and febrile neutropenia" which explores the role of procalcitonin (PCT) as a valuable biomarker for guiding antibiotic therapy in patients with cancer and febrile neutropenia. The manuscript describes a prospective randomized study about antibiotic de-escalation practices and PCT monitoring in cancer patients.

The manuscript is well-structured, providing a clear background, proper methodology, statistical analysis, and ample discussion.

The authors included tables and figures that effectively give a summary of the obtained results.

The conclusion appropriately summarizes the key points.

The authors wrote a meticulous structured manuscript with a clear and logical flow.

In my opinion, the study has a relatively small sample size (89 patients) which limits the generalizability of the study' findings. Therefore, I suggest that the authors would include study limitations and further researches.

I therefore recommend that the paper can be published after minor revisions.

Author Response

I reviewed the manuscript entitled "Procalcitonin level monitoring in antibiotic de-escalation and stewardship program in patients with cancer and febrile neutropenia" which explores the role of procalcitonin (PCT) as a valuable biomarker for guiding antibiotic therapy in patients with cancer and febrile neutropenia. The manuscript describes a prospective randomized study about antibiotic de-escalation practices and PCT monitoring in cancer patients.

The manuscript is well-structured, providing a clear background, proper methodology, statistical analysis, and ample discussion.

The authors included tables and figures that effectively give a summary of the obtained results.

The conclusion appropriately summarizes the key points.

The authors wrote a meticulous structured manuscript with a clear and logical flow.

In my opinion, the study has a relatively small sample size (89 patients) which limits the generalizability of the study' findings. Therefore, I suggest that the authors would include study limitations and further researches.

I therefore recommend that the paper can be published after minor revisions.

Response: Thank you for your positive feedback and your constructive comments. We agree with your observation on the limitations regarding the study's sample size and its potential effect on the generalizability of our findings. This limitation has been acknowledged in the discussion section. In response to your suggestion, we have also revised the manuscript to include recommendations for future research.

We appreciate your recommendation for publication following minor revisions and hope that our revisions now meet your expectations.

Reviewer 5 Report

Comments and Suggestions for Authors

Thank you for the opportunity to review this paper. I believe this article can consolidate what is already known about the use of procalcitonin in the setting of cancer patients. 

Overall, I think it is a well-structured study, and I have only minor comments:

1- I cannot understand why a lower cut-off than the literature data (30% vs 80%) was chosen to define PCT drop. In line 154 you explain that you made this decision because, in cancer patients, there may be higher than average procalcitonin values due to the underlying disease. As much as I agree with that statement, the value of “30%” seems a bit arbitrary to me (is there data in the literature comparing PCT values in cancer patients vs. the general population to establish how to determine that value?)
2- Was the clinician's decision to de-escalate antibiotic therapy motivated ONLY or MAINLY by clinical conditions? As a clinician, I am aware that procalcitonin influences decisions about de-escalation/interruption of antibiotic therapy. So if, as I believe, physicians were aware of their patients' procalcitonin values, I think a bias is created in the presentation of the data (physicians de-escalated/interrupted antibiotic therapy ALSO because procalcitonin was reduced, and in Tab 2 it is said that, in those patients, procalcitonin was significantly lower than in the control group)

Author Response

Thank you for the opportunity to review this paper. I believe this article can consolidate what is already known about the use of procalcitonin in the setting of cancer patients.

Overall, I think it is a well-structured study, and I have only minor comments:

1- I cannot understand why a lower cut-off than the literature data (30% vs 80%) was chosen to define PCT drop. In line 154 you explain that you made this decision because, in cancer patients, there may be higher than average procalcitonin values due to the underlying disease. As much as I agree with that statement, the value of “30%” seems a bit arbitrary to me (is there data in the literature comparing PCT values in cancer patients vs. the general population to establish how to determine that value?)

2- Was the clinician's decision to de-escalate antibiotic therapy motivated ONLY or MAINLY by clinical conditions? As a clinician, I am aware that procalcitonin influences decisions about de-escalation/interruption of antibiotic therapy. So if, as I believe, physicians were aware of their patients' procalcitonin values, I think a bias is created in the presentation of the data (physicians de-escalated/interrupted antibiotic therapy ALSO because procalcitonin was reduced, and in Tab 2 it is said that, in those patients, procalcitonin was significantly lower than in the control group)

Response: Thank you for your thoughtful review and valuable comments. We appreciate your positive feedback on the structure of the study and the potential contribution of our findings to existing knowledge on PCT use in cancer patients.

1- Regarding the cut-off value for PCT drop, we agree that the choice of 30% might seem arbitrary. While there is limited data in the literature specifically comparing procalcitonin levels in cancer patients versus the general population, we based our decision on studies indicating that cancer patients may present with elevated baseline procalcitonin due to factors unrelated to infection such as their underlying cancer (reference 22). Hence, in cancer patients, PCT may not drop by 80%, as it may remain elevated due to their underlying cancer, unlike in non-cancer patients where the PCT may be elevated due to infection solely, and we may observe a bigger drop in PCT. Furthermore, our choice of 30% was supported by another study that showed that, in patients with cancer who have bacterial infections, a decrease in PCT level has been shown to reflect the response to antimicrobial therapy, with a significant decline in median PCT level of 30% in patients who responded to therapy (from 0.27 ng/ml to 0.19 ng/ml; p=0.002). Please refer to reference #23 in the revised manuscript (Al Shuaibi, M.; Bahu, R.R.; Chaftari, A.M.; Al Wohoush, I.; Shomali, W.; Jiang, Y.; Debiane, L.; Raad, S.; Jabbour, J.; Al Akhrass, F.; et al. Pro-adrenomedullin as a novel biomarker for predicting infections and response to antimicrobials in febrile patients with hematologic malignancies. Clin Infect Dis 2013, 56, 943-950, doi:10.1093/cid/cis1029). 

However, we acknowledge in the discussion section that this area requires further research, and we have clarified this point in the manuscript to address potential concerns about the chosen threshold.

2- The clinician's decision to de-escalate antibiotic therapy was motivated MAINLY by clinical conditions. However, we agree with the reviewer that, while clinical conditions played a primary role, the clinician's awareness of PCT levels may have influenced the decision to de-escalate or interrupt antibiotic treatment.

We have acknowledged this potential bias and have revised the manuscript accordingly.

We would like to extend our sincere thanks to all the reviewers for their valuable time, insightful comments, and constructive suggestions. Your feedback has been instrumental in improving the quality of this manuscript. We hope that you find our revision satisfactory and the manuscript suitable for publication in your prestigious journal.

Sincerely,

Anne-Marie Chaftari
